# Impact of Obesity on the Course of Management of Inflammatory Bowel Disease—A Review

**DOI:** 10.3390/nu14193983

**Published:** 2022-09-25

**Authors:** Agata Michalak, Beata Kasztelan-Szczerbińska, Halina Cichoż-Lach

**Affiliations:** Department of Gastroenterology, Medical University of Lublin, Jaczewski St 8, 20-954 Lublin, Poland

**Keywords:** inflammatory bowel disease, obesity, adipose tissue, inflammation, treatment

## Abstract

It is already well-known that visceral adipose tissue is inseparably related to the pathogenesis, activity, and general outcome of inflammatory bowel disease (IBD). We are getting closer and closer to the molecular background of this loop, finding certain relationships between activated mesenteric tissue and inflammation within the lumen of the gastrointestinal tract. Recently, relatively new data have been uncovered, indicating a direct impact of body fat on the pattern of pharmacological treatment in the course of IBD. On the other hand, ileal and colonic types of Crohn’s disease and ulcerative colitis appear to be more diversified than it was thought in the past. However, the question arises whether at this stage we are able to translate this knowledge into the practical management of IBD patients or we are still exploring the scientific background of this pathology, having no specific tools to be used directly in patients. Our review explores IBD in the context of obesity and associated disorders, focusing on adipokines, creeping fat, and possible relationships between these disorders and the treatment of IBD patients.

## 1. Introduction

Inflammatory bowel disease (IBD) constitutes a well-known example of pathology with a remitting–relapsing course. Great progress towards understanding its pathology, highly specialized treatment, and meeting the patients’ needs has been made in the last few years. Simultaneously, a great effort to better understand the social, clinical, and psychological profile of a single IBD patient has been made, as well. Nevertheless, unresolved issues independently connected with the course of IBD and its potential management strategies still require being clarified. Among such phenomena, the idea of obesity plays an important role in the context of IBD [1,2,3,4]. The general prevalence of obesity among IBD patients is quite diversified and seems to be similar in comparison to healthy subjects, taking into consideration the majority of already conducted studies [5,6,7,8]. The exact number of cases and the commonness of IBD increase all over the world, especially in developing countries. Approximately 15–40% of the adults diagnosed with ulcerative colitis (UC) and Crohn’s disease (CD) suffer from obesity (BMI ≥ 30 kg/m^2^); an additional 20–40% are overweight. Body mass disturbances are noted to a similar extent in both UC and CD. The corresponding tendencies are observed in the population of teenagers with IBD. Obesity can also be linked to a higher risk of development of CD; however, this association seems to be unrelated to UC [9,10,11]. Moreover, according to the Danish National Birth Cohort (counting more than 75 thousand women), obesity before getting pregnant was connected with a 1.9-fold increased risk of CD (but not UC) in the examined female population [12]. Another cohort of examined women presented a similar pattern of results [13]. Furthermore, Yan et al. proved that maternal obesity constitutes a milieu that favors the increased expression of proinflammatory cytokines, promoting the development of inflammation within the large intestine of sheep fetuses and offspring [14]. Additionally, another study showed that maternal high-fat diet increases the risk of IBD in mouse offspring [15]. It can be assumed that maternal obesity is linked to an increased risk of CD in both the mother and her offspring. On the other hand, regular physical activity is believed to lower the risk of IBD. The systematic review of population-based studies done by Ng et al. suggested that the prevalence of IBD increases, especially in developing countries, with societies adopting the so-called Western model of life. The greatest numbers of new IBD cases are found in Europe (UC diagnosed in 505 per 100,000 people in Norway; CD diagnosed in 322 per 100,000 people in Germany) and North America (UC—286 cases per 100,000 people in the USA; CD—319 cases per 100,000 people in Canada). The total prevalence of IBD exceeded 0.3% in North America, Oceania, and many countries in Europe; therefore, the problem is still emerging [16,17,18,19]. A major characteristic pattern describing the Western model of life involves a wide variety of environmental factors which appear to be inseparably related to the development of IBD *de novo*. They include dietary modifications, changes in the microbiome, improved status of hygiene, and increased use of antibiotics. The dietary revolution which has occurred in modern societies is connected with the domination of modified foods in everyday life, animal-derived products being more common than plant-based diets, high levels of consumption of simple sugars, and, in general, a significantly increased consumption of calories. Even though scientific data suggest that dietary factors might constitute a direct releasing factor for the development of IBD, it must be highlighted that the abovementioned dietary changes result in the overall increased risk of obesity in societies around the world. Obesity has turned out to be an increasing global burden in recent decades and, together with the still rising number of new IBD cases in developing countries, it should be perceived as a risk factor for IBD. The observations from clinical research aiming to determine the actual role of obesity in the premorbid period of IBD are quite ambiguous [20,21,22,23]. One of the large cohort studies suggested that obesity present in the early stage of adolescence was positively associated with an early diagnosis of CD, before the age of 30, but inversely correlated with a future diagnosis of UC at any age [10]. On the contrary, the report of the European Prospective Investigation into Cancer and Nutrition did not indicate any relationship between the BMI and the future risk of development of CD or UC [24]. The data appear to be quite obscure; nevertheless, the association between the body mass status and IBD is undeniable. A significant role in this area is played by adipose tissue and adipokines released by this organ [25,26,27,28]. Thus, we decided to perform a review of available literature on the involvement of obesity and adipose tissue in the modulation of metabolic and inflammatory background in IBD, giving special attention to their role in the treatment and general management of affected patients.

## 2. IBD and Its Metabolic Background—A Relationship Not to Be Neglected

Despite the complex etiology of IBD, its involvement in the spectrum of metabolic disorders constitutes another crucial dependency. The concept of inflammation accompanying obesity, insulin resistance, and dyslipidemia appears to create this pathophysiological linking element between them. Finally, these systemic metabolic consequences might directly modify a microbial environment in IBD subjects. The data on the relationship between metabolic syndrome and IBD are somewhat unclear [29]. Multiple reports indicate a higher prevalence of the coexistence of these disorders, especially concerning UC and metabolic syndrome [30,31,32]. Nevertheless, other observations showed a similar presence of a metabolic syndrome in the course of IBD compared to the healthy population. Possibly, the multifactorial character of the metabolic syndrome complicates obtaining a distinct answer to this issue [33,34]. A potential association between IBD and diabetes requires further studies as well; however, sharing common genetic variants was raised as a link between UC and diabetes [35]. Moreover, the observation of comorbidity of IBD and type 2 diabetes in former surveys highlighted intestinal dysbiosis and altered epithelial barrier as possible causative factors [29]. Insulin resistance was also noted to occur more often in UC patients [36,37]. Coincidence of metabolic-associated liver disease (MAFLD) is another phenomenon postulated to be more frequent among IBD patients. Hypertransaminasemia observed in them is predominantly believed to be associated with liver steatosis due to obesity or intake of steroids or immunosuppressants. Because of the composed background of both IBD and MAFLD and the common factors involved in their development (e.g., chronic inflammation, microbiome alterations), their relationship has not been fully clarified so far. Nevertheless, recent observations have proven the presence of a relationship between IBD and both MAFLD and liver fibrosis regardless of the involved risk factors [38,39,40,41,42]. Translation of this knowledge into the practical attitude to the treatment and general management of this group of patients seems to be of great importance.

## 3. Adipose Tissue as an Endocrine Organ Involved in Metabolic and Inflammatory Pathologies

Even the previously performed surveys on the role of adipose tissue revealed that, except for storing energy, it participates in essential endocrine functions [43,44]. Adipose tissue, explored all the time, modulates human metabolism in the auto-, para-, or endocrine pattern via secretion of biologically active molecules: hormones, peptides, and various specific cytokines (adipokines). There is a growing body of evidence that adipokines participate in numerous immunological and inflammatory reactions related to multifactorial pathologies [45,46,47,48]. Nowadays, obesity is described more and more frequently as a low-grade inflammatory state perceived as a triggering factor of numerous diverse chronic conditions. Historically, adipose tissue constituted a neutral store for excessive fat in the organism. Currently, it is well-established that adipose tissue plays a role of an endocrine gland, secreting a certain type of cytokines—adipokines, exerting a proinflammatory (TNF-α, IL-6) or anti-inflammatory role (adiponectin) [49]. Of note, each form of obesity is characterized by a unique biochemical profile. The presence of *creeping fat* (abdominal fat migrating to the wall of the inflamed small intestine) in CD patients has been proven to create a specific environment for cytokines in comparison to splanchnic or subcutaneous storage of fat. As a result, a hypothesis was proposed that the overexpression of various proinflammatory molecules in the course of IBD might provoke the underlying disease-specific inflammation [50]. Figure 1 presents the general relationships between obesity, adipokines, and IBD. 

## 4. Apelin—Epithelium and Lymphatic Vessels in the Scope of IBD

Investigations conducted on colonic tissue in mice and humans diagnosed with colitis/IBD revealed an increased concentration of apelin [51,52]. In previous studies, apelin was found to be expressed in the greatest concentrations within epithelial cells. Of note, adding the synthetic apelin to the cell culture turned out to stimulate the proliferation of epithelial cells [51]. A relatively new discovery concerns the participation of apelin in the stabilization of lymphatic vessels. This association can not be neglected as the impaired lymphatic transport of the mesentery was described before as a potential causative factor in the course of CD. Moving further, mesenteric tissue in CD patients was presented as a crucial source of apelin. Interestingly, the Il10−/− mouse model with confirmed colitis given apelin was observed to manifest a decreased production of proinflammatory molecules (e.g., TNF-α, IL-6, and IL-1β), reduced severity of inflammation, and, finally, improved intestinal lymphatic functions, assessed due to a greater lymphatic vessel density and confirmed by enhanced lymphatic drainage evaluated with lymphangiography. In general, the properties of apelin might be described as having regenerative abilities on intestinal cells and supportive features concerning lymphatic drainage [53,54].

## 5. Leptin and Its Pleiotropic Function

Leptin belongs to one of the best known so far and the most important adipokines released by adipose tissue. It is secreted almost exclusively by adipocytes in a settled circadian rhythm—its level increases up to 30% at night [55]. Food intake and hormonal balance (e.g., related to insulin) constitute the major regulating factors of leptin production. Its synthesis is also dependent on sex hormones: testosterone inhibits and ovarian hormones promote the secretion of leptin [56,57,58]. Consequently, the concentration of this adipokine differs between genders regardless of age and body weight. Leptin is widely known to serve the role of a central inhibitor of hunger and the agent stimulating the use of energy. Its decreased level in the bloodstream increases hunger and mediates the preservation of energy resources through neurohormonal and behavioral changes to preserve the basic vital processes [59,60,61,62,63]. Thus, in case of starvation or reduction of the amount of adipose tissue, the concentration of leptin decreases, which is followed by the reduced use of energy—everything to maintain the function of crucial organs: the brain, heart, and liver [64,65,66,67]. A high level of leptin commonly observed in obesity is probably inseparably related to underlying insulin resistance, metabolic syndrome, and chronic mild inflammation [68,69]. The recent years have given a new scope on the fundamental role of leptin in the regulation of CD4^+^CD25^+^ Treg proliferation, which might explain its involvement in the course of autoimmune diseases [70]. Notably, Treg lymphocytes constitute an essential source of leptin, simultaneously secreting and presenting receptors for this adipokine [71]. This proves that leptin may take part in the negative feedback controlling the level of Treg. Leptin (circulating in the bloodstream and released locally) binds receptors of CD4^+^CD25^+^ Treg and effector T cells CD4^+^CD25^−^ [72]. It causes differentiated consequences. CD4^+^CD25^+^ Treg cells become hyperactive with a tendency to anergy, while among T effector cells, leptin promotes polarization of Th1 (leading to the increased expression of IFN-γ and the decreased expression of IL-4) and secretion of proinflammatory cytokines (TNF-α, IL-2, IL-6), which closes the loop and stimulates further production of leptin, synergistically enhancing its action [73]. De Rosa et al. pointed out that neutralization of leptin reverses the anergy and restores the autoimmune response dependent on Treg lymphocytes [70]. The expression of receptors for leptin is multiplied by various mediators of inflammation. Acute infection and sepsis were shown to stimulate the synthesis of leptin [74,75,76,77]. All the abovementioned issues might be related to the potential involvement of leptin in the course of IBD due to its participation in the phenomenon of inflammation according to numerous molecular paths [78,79,80]. Therefore, leptin seems to play a significant role in the idea of IBD. It was found to be involved in the induction of lipid droplet formation within intestinal epithelial cells. In the discussed study, simultaneously, the enhanced generation of CXCL1/CINC-1, CCL2/MCP-1, and TGF-β was noticed as well. With the correspondence to previous research, leptin stimulated cell proliferation. The administration of its pegylated form antagonist (PG-MLA) also helped in avoiding experimental chronic colitis. This anti-inflammatory action was reflected by a lowered concentration of cytokines together with an increase in mucosal Treg cells. The phenomenon of lipid droplet induction and the accompanying reactions were based on the mammalian target of rapamycin (mTOR) pathway; it was completely inactive in the presence of rapamycin. According to the already known data, leptin, similarly to adiponectin, directly modulates the function of intestinal epithelial cells and participates in the maintenance of intestinal homeostasis [81]. Its concentration was found to be elevated in the majority of investigations conducted on UC and CD patients compared to the controls. Leptin was associated positively with the level of proinflammatory cytokines, too (IL-1, TNF-α), and with the endoscopic activity of the disease in some cases. Of note, its overexpression was demonstrated in mesenteric adipose tissue of CD and UC patients, too [82,83,84,85,86,87]. Nevertheless, some of the available investigations conducted on the concentration of leptin in subjects with IBD revealed its lowered concentration or even showed the lack of the association between CD and the concentration of leptin [88,89]. According to the already obtained data, the main action of leptin is directly possible due to its modulatory effect exerted on T cells. The administration of leptin antagonists in an experimental model of colitis promoted the recruitment of Treg cells, decreasing the colonic infiltration by proinflammatory cytokines [90,91].

## 6. Adiponectin in the Setting of Metabolic and Inflammatory Disorders

Adiponectin (Acrp30) is a subsequent representative of cytokines produced mainly by mature adipocytes [92,93,94]. A lot of already performed surveys indicate that adiponectin mediates the secretion of cytokines by monocytes, macrophages, and dendritic cells (increased production of IL-10 and decreased production of IFN-γ), promoting maintenance of the anti-inflammatory milieu [95,96,97,98]. Its anti-inflammatory role has been highlighted in the course of atherosclerosis, among others. Furthermore, proinflammatory molecules, such as TNF-α and IL-6, lower the expression of Acrp30 [99]. Of note, it has been proven that its relatively low level correlates with the progression and severity of both metabolic-associated fatty liver disease (MAFLD) in general and following steatohepatitis [100]. Similarly, obesity and diabetes mellitus type 2, are other states accompanied by a decreased concentration of adiponectin [101]. The current point of view on central abdominal obesity relates it to chronic underlying inflammation, which inhibits the synthesis of Acrp30, leading to the development of hypertension, atherosclerosis, diabetes, and cancer [102,103,104,105]. In the scope of the inflammation concerning adipose tissue, metabolic syndrome and IBD can be perceived as the associated pathologies with a common basic feature—disrupted secretion of adipokines in their course [106,107,108,109,110]. Nevertheless, this imbalance is more severe in subjects with metabolic syndrome because of the involvement of the whole-body adipose tissue. Thus, IBD and metabolic disorders should be discussed and explored together. A decreased concentration of omentin 1 in both conditions confirms this point of view [29,111]. Adiponectin, obesity, and inflammation appear to possess mutual direct interference. From this perspective, the aforementioned interactions are worth emphasizing in order to better understand the potential involvement of adiponectin in the natural history of IBD. Thus, the axis related to adiponectin might concern the idea of inflammation in numerous ways, according to the certain disorder. The issue of being a potential predisposing or protective factor in the course of inflammatory pathologies still requires further studies of Acrp30. One of the surveys performed to assess the concentration of adiponectin in the creeping fat of CD patients revealed its higher concentration compared to the non-creeping fat in the course of CD, the fat obtained from UC patients, and, finally, from the controls [112]. A model of adiponectin deficiency was applied in two groups of experimental colitis mice (induced by dextran sulfate sodium (DSS) and trinitrobenzene sulphonic acid (TNBS)), according to the study conducted by Fayad et al. The external stimulation of DSS-exposed mice with adiponectin caused the elevation in the synthesis of IL-6 and macrophage inflammatory protein-2 (MIP-2); it was not observed in the situation of adiponectin knockout [113]. Nishihara et al. followed a similar manner of investigation on adiponectin-deficient mice in DSS- and TNBS-induced colitis. In that case, the administration of adiponectin had an anti-inflammatory impact on colonic epithelial cells. The presented discrepancies between the mentioned surveys are believed to be caused by different types of knockout mouse models and adiponectin forms that were applied [114]. Another in vitro investigation performed on colonic epithelial cell line HT-29 concerned the effect of globular and full-length adiponectin. Its first type was especially connected with pro-proliferative and proinflammatory features, reflected by the stimulation of extracellular signal-regulated kinase (ERK) and p38 mitogen-activated protein kinase (MAPK) NF-κB signaling acting directly with colonic epithelial cells [115]. It is also worth mentioning a potential influence of the specific fat-conditioned milieu in IBD-derived patients. After being exposed to it, the human NCM60 epithelial cell line presented a lowered expression of adiponectin receptor 1 (AdipoR1). The effect of this down-expression was followed by the deterioration of colitis in TNBS-induced colitis mice [116]. Another former observation obtained from the investigation of adiponectin-deficient mice administered DSS showed the exacerbation of underlying colitis with an additional release of inflammatory molecules (IL-1β, IL-4, and IL-6) associated with a more pronounced activation of B cells and enhanced STAT3 signaling within the colon. The epithelium of the examined objects was noticed to present a lowered ratio of cell proliferation together with increased apoptosis and accompanying cellular stress. Due to in vitro surveys, these consequences could be avoided with the use of adiponectin. All these data support the concept that adiponectin maintains the intestinal balance [117].

## 7. Resistin—A Significant Marker of Systemic Inflammation

The year 2001 appears to be the first time when resistin was described as a peptide released by adipocytes, responsible for the development of insulin resistance [118]. Further surveys revealed that in humans resistin is synthesized mainly by monocytes and macrophages within adipose tissue and peripheral organs (spleen, bone marrow), in contrast to animal models [119,120,121,122,123]. Nevertheless, the already collected data concerning its features and functions are still scanty. Resistin shares a lot of common characteristics with proinflammatory cytokines and activating transcriptional factor NF-kβ. An increased expression of this factor was observed in the process of differentiation among monocytes and macrophages and after the stimulation with TNF-α, IL-1β, IL-6, and LPS [124,125,126,127,128,129,130]. The receptor for resistin still has not been found, therefore, the details concerning its signalization pathways remain not fully elucidated [131]. There is a demand for new studies focusing on the direct role of resistin in the pathogenesis of insulin resistance and its interactions with the cells of the immune system [132]. Even though resistin has been explored in numerous systemic disorders (e.g., atherosclerosis, insulin resistance), its potential clear role in the course of IBD has not been elucidated so far. Nonetheless, higher levels of resistin are known to be associated with IBD [88,133] and tend to decrease during therapy with infliximab [134]. Furthermore, there have been attempts to correlate the concentration of resistin in stool with the prediction of a flare in the course of pediatric IBD. The results of the already performed studies are quite promising, showing a relatively high diagnostic accuracy of resistin in this area. Other observations revealed significantly higher concentrations of resistin in pediatric IBD patients compared to the controls. What is more, a positive association between serum resistin and fecal calprotectin together with both CRP and white blood cell count has been noted as well [135,136].

## 8. Chemerin—A Powerful Chemoattractant

The innate immune system and its modulating particles have been described as the target for the activity of chemerin, while it behaves as a chemoattractive agent for immune cells [137,138,139]. A single investigation conducted on CD and UC patients revealed an elevated level of chemerin and a simultaneously decreased concentration of adiponectin [140,141]. Another in vivo observation study based on the external administration of chemerin to mice with DSS-induced colitis resulted in the exacerbation of inflammation due to a noticed enhanced secretion of TNF-α and IL-6 with an accompanying lowered number of anti-inflammatory macrophages synthesizing IL-10. These investigations were also confirmed during in vitro analyses. One of them showed that deficiency in the receptor for chemerin (chemokine-like receptor 1 (CMKLR1)) can be followed by the development of DSS-induced colitis in a delayed period of time [142]. A recent observation demonstrated that epithelial chemerin–CMKLR1 signaling is dependent on lactoperoxidase and its administration during microbiota-driven neutrophilic colitis could potentially improve microbiota dysbiosis, giving a perspective for such a treatment in IBD [143]. Additionally, another survey performed in the situation of DSS-induced colitis showed that in vitro presence of chemerin made it impossible for macrophages to create an anti-inflammatory phenotype, which led to a deteriorated expression of arginase-1 and IL-10 [144].

## 9. Other Adipokines

Visfatin constitutes another worth-mentioning adipokine, correlating significantly with the concentration of visceral fat and mesenteric adipose tissue. It was found to participate in the intra- and extracellular processes related to obesity [145,146]. Its higher serum expression was observed in the general population of IBD patients and their colonic samples [147,148,149]. Among pediatric IBD patients, visfatin correlated positively even with a disease activity assessed by biopsies [150]. Of note, vaspin, a newly discovered adipokine, has been proven to be characterized by insulin-sensitizing and anti-inflammatory effects. Some investigations have revealed its higher concentration in the course of IBD. Its expression was even observed among adipocytes of the mesenteric adipose tissue (MAT) in IBD patients [151,152,153]. Omentin-1 (known as intelectin-1) is worth mentioning, too. It participates in the maintenance of insulin sensitivity and body metabolism. It is also known for its anti-inflammatory features. A lowered level of omentin-1 was detected in the general population of IBD patients compared to the controls [154]. Except for its decreased value in colonic tissue, it was also shown to correlate with CD severity [155]. Meteorin-like constitutes a representative of adipo-myokines and was observed to be highly expressed in white adipose tissue (WAT). The discussed peptide appears to play an anti-inflammatory role by increasing beige fat thermogenesis [156,157]. WAT of CD patients turned out to contain a greater concentration of meteorin-like compared to healthy subjects [158]. Gastric endocrine cells are the source of ghrelin which decreases the ratio of transformation of preadipocytes into their mature forms via the inhibition of PPAR-γ. Ghrelin attenuates leptin-induced proinflammatory responses in macrophages and T cells, causing a reduction in the concentration of proinflammatory cytokines (including TNF-α, IL-1, IL-6, and IL-8) and lowering the secretion of leptin in the gastrointestinal tract [159]. Expression of ghrelin in the course of IBD turned out to be significantly higher in comparison to the controls [160,161]. In a model of TNBS-induced murine colitis, intraperitoneal administration of exogenous ghrelin improved the general outcome without an impact on PPAR-γ expression [162].

## 10. IBD and Adipose Tissue—Take It or Leave It?

It is all about fat. Obesity must be perceived as the situation of permanent low-grade inflammation with activation of various signaling pathways in its course mediated via, e.g., CRP and NF-κB. The role of adipose tissue definitely exceeds storage features as it is a source of the abovementioned wide spectrum of adipokines. Thus, it is involved in metabolic homeostasis and immune functions [163]. The general idea of adipose tissue has definitely changed in recent years. From the organ characterized by its endocrine and metabolic properties, the perception of adipose tissue raised and evolved to the modulator of inflammation, linking fat with CD, for instance [164]. Observation studies conducted in the 1990s promoted this way of thinking, broadening and irreversibly changing the concept of adipose tissue [165,166]. Everything turned out to be mediated via adipokines. Except for their general metabolic and immunological functions in humans, their modulatory properties are reflected by hyperplasia of mesenteric fat that, in turn, is creeping around the inflamed loops of the small intestine.

## 11. General Idea of Mesenteric Adipose Tissue

MAT hypertrophy does not belong to novel findings; its concept was initially proposed a long time ago, in 1932, by B.B. Crohn, as an underlying feature of CD [167]. Bowel inflammation in this particular situation involves the surrounding adipose tissue along the mesentery. MAT ranges from the mesenteric attachment to the root of the mesentery [168]. This hypertrophic kind of mesenteric fat attached to the inflamed bowel is well-known as *creeping fat* or *fat wrapping.* Creeping fat is a phenomenon pathognomonic for CD. It contains adipocytes, endothelial cells, immune cells, fibroblasts, preadipocytes, and stem cells. This type of activated adipose tissue produces a wide range of mediators, e.g., cytokines, adipokines, fatty acids, and growth factors. Creeping fat is settled between the serosa and the muscularis propria, which indicates that adipocytes remain in direct contact with intestinal smooth muscle cells. Continuing this theory, the presence of creeping fat involves muscularis propria hyperplasia, transmural inflammation, and, finally, intestinal fibrosis. These phenomena might finally lead to the development of the stricturing form of CD [169,170].

## 12. Adipose Tissue and IBD—Where Is the Exact Beginning?

Interestingly, some data indicate that MAT can exert a protective action due to the improvement of local host defense by stimulating local inflammation, which can even reduce the risk of perforation [171]. According to this point of view, the expression of leptin (enhancing the expression of toll-like receptors (TLRs) in preadipocytes and their mature forms) has been shown to be upregulated locally, but not systemically in the course of IBD [89,172,173]. Nevertheless, in which specific way can the adipose tissue become the main causative axis of inflammation in the course of CD? The events occurring within the mesenteric fat tissue appear to be described as the core of the CD initiation. This cascade is hard to be clearly explained, even though the transmural inflammation together with an accompanying chronic bacterial translocation constitute its background. Of note, former data indicate that adipocytes, as well as preadipocytes, express certain receptors belonging to the innate immune system, such as TLRs and nucleotide oligomerization domains 1 (NOD1) and 2 (NOD2). In the corresponding mouse models, the Myd88−/− type presented enhanced production of proinflammatory particles in the course of chronic DSS-induced colitis. Therefore, mice with an impaired innate immune system did not manage to respond to these penetrating bacteria, having a remarkably higher mortality; this may suggest that the mesenteric fat plays the role of a possible second barrier line. Furthermore, the concentration of leptin within mesenteric fat was elevated in comparison to TLR9−/− mice. On the other hand, IL-6 production was not influenced by the lack of TLR9 signaling that other innate receptors contribute to the abovementioned molecular changes [89,171,172,173,174].

## 13. Adipose Tissue-Related Implications in the Clinics of IBD

Adipose tissue and the bowel interact with each other. Therefore, obesity can be perceived as a composed phenomenon directly connected with inflammation, the possibility of increased permeability of the walls of the intestines and dysbiosis as the consequence. Meanwhile, inflammatory markers produced by adipose tissue constitute a parallel pathology triggering morphological dysfunction of the gastrointestinal tract. This kind of self-perpetuating loop with its consequences can be observed in the natural history of IBD [175]. Considering IBD, a significant role of malnutrition in its course must be emphasized, together with possible sarcopenia in the consequence. The loss of lean muscle coexisting with decreased muscle strength can be observed in IBD patients due to the loss of nutrients or an increased demand for them as well. Similarly, self-introduced diet restrictions can be involved as well, present in IBD patients usually to avoid a potential recurrence. Sarcopenia constitutes a well-known factor responsible for the poor clinical outcome in IBD (e.g., higher numbers of required surgeries and rehospitalizations). Another point that must be raised here is the phenomenon of sarcopenic obesity in IBD patients. Thus, a formally adequate BMI might be the result of more body fat and low muscle mass. In this context, the diagnostic utility of the BMI in this kind of disorders can be limited. Of note, sarcopenic obesity has been proven to correlate with a higher rate of rehospitalizations in the case of a flare. It gives another point of view on the idea of body mass in IBD patients. From this perspective, the assessment of nutritional status among these people should be obtained with more detailed procedures, e.g., bioelectrical impedance analysis [176,177,178,179,180,181].

## 14. T Cells, Creeping Fat and Here the Story Goes

The character of resident T lymphocytes in IBD patients is modified due to the transition from the regulatory environment comprised of M2 macrophages, NK cells, and CD4+ Treg cells into the proinflammatory milieu characterized by the presence of macrophages M1, CD4+ Th1, and cytotoxic subtypes of CD8+. Being a member of the IL-6 family, leptin is produced by adipocytes proportionally to the mass of adipose tissue. Leptin induces the synthesis of proinflammatory cytokines in monocytes and participates in the modulation of the immune response via T lymphocytes. Adiponectin produced in the adipose tissue in a reverse manner according to its amount sensitizes the cells to insulin and behaves as an anti-inflammatory cytokine in objects without IBD. Nevertheless, in IBD patients, especially in the course of CD, adiponectin turns out to constitute a proinflammatory particle, leading to the increased proliferation of epithelial cells and enhanced release of proinflammatory cytokines. Macrophages and monocytes are known sources of resistin; their expression is induced by IL-1, IL-6, and TNF-α, and the concentration of these molecules was found to correlate with the level of markers of the acute phase (e.g., CRP). Moreover, resistin might promote the synthesis of IL-6 and TNF-α. The investigations conducted in recent years revealed the presence of a loop with positive feedback between the adipose tissue, the colon, and the mucous membrane. Namely, IL-17A is released by adipocytes stimulated by P-substance and, simultaneously, the mucous membrane of the colon presents an increased concentration of receptors for IL-17A. Obesity and a high-fat diet, together with paramural inflammation in CD, are factors responsible for the disabled function of the mucosal barrier through the modification of proteins involved in tight junctions [182,183,184,185]. Preadipocytes and adipocytes present on their surface functional pattern recognition receptors (e.g., TLRs and NODs), which interact with bacteria-derived molecules by releasing proinflammatory molecules. Finally, the translocation of bacteria to the lumen of the intestine through a weakened epithelial barrier becomes possible. The idea of creeping fat is an already mentioned and well-known phenomenon in CD patients; however, it still requires detailed studies. It is formed by hyperplastic adipose tissue surrounding inflamed loops of the intestine [186,187,188,189]. Kredel et al. revealed the presence of quite a unique character of ileal mesenteric tissue in CD patients with features of adipocyte hyperplasia and dense T cell infiltration together with fibrosis. The examined colonic fat obtained from CD and UC patients investigated in the discussed survey did not present such abnormalities. Interestingly, an inverse relationship between disease severity and the amount of pro- and anti-inflammatory T cell subsets was observed in CD after comparing ileal and colonic fat. Such a finding constitutes proof of the great heterogeneity of IBD. Consequently, even ileal and colonic types of CD, according to genetic studies, have been presented as two independent forms of the disease. The background of this phenomenon might be explained by the involvement of NOD2 variants in the disease of the small intestine and the participation of HLA alleles in colonic CD [190]. Going further, ileal CD, colonic CD, and UC appear to be different entities from a genetic point of view. The location of IBD is also inseparably connected with a certain gut microbiota composition. The aforementioned aspects of IBD emphasize a need for an individual attitude to each case together with searching for a more personalized character of the treatment in the future [191,192].

## 15. IBD, Adipose Tissue, and Pharmacological Treatment—Associations and Speculations

How exactly should we translate this molecular knowledge based on the inflammatory phenomena related to fat and obesity into the everyday clinical practice concerning IBD patients? According to the latest data, adipose tissue can be perceived as a key player in the development of IBD and a general course of the disease, especially in the context of CD. MAT is perceived as the source of a reactive immunological zone around the inflamed intestine, creating a milieu which could constitute a target for direct treatment. Even though a molecular background of creeping fat in CD patients still remains the issue that has not been fully elucidated, the involvement of visceral adipose fat (VAT) in the course of CD-related inflammation is undeniable, indicating once again a possible target for the treatment and monitoring of the severity of the disease. For that reason, attempts to evaluate IBD activity with the use of the measurement of MAT/VAT volumes or MAT-derived mediators seem to be completely reasonable and worth exploring. Indeed, it is not such a new and revolutionary concept [174,193,194]. The year 1992 seems to be the first time when the phenomenon of fat wrapping in the intestines was shown to be correlated with ulceration, stricture formation, increased wall thickness, and transmural inflammation in CD by Sheehan et al. [195]. Since that finding, subsequent investigations have been performed in this area, providing, in general, corresponding results. Mesenteric fat assessed with CT images has significantly reflected the activity of CD quantified with CRP and CDAI. Going further, the mesenteric fat index (MFI; visceral-to-subcutaneous fat) was proposed to be perceived as an indicator of complicated CD. Consequently, visceral deposition of fat was noted in patients with fistulizing and stenotic disease [196,197]. VAT/total fat mass (FM) ratio has also been examined in the context of CD severity; its higher values corresponded with B2 and B3 CD behavior, based on the Montreal classification. It has also been proven to have predictive value, being related to the shorter period of remission in females with CD. In addition, the analysis performed on 482 CD patients from the PRISM database similarly showed a relevant dependency between the amount of visceral fat and the risk of developing penetrating disease and the need for surgical treatment [198]. The pediatric population of CD patients presented a similar tendency as well; the survey by Uko et al. demonstrated a greater risk of fistulizing and fibrostenotic form of the disease in the case of higher VAT volumes (evaluated with CT) in them, together with an increased number of hospitalizations and higher disease activity scores at the baseline. On the other hand, the measurement of intra-abdominal adipose tissue with MR indicated its relationship with a more complex and prolonged character of the active disease. Even postoperative morbidity after bowel resection due to CD turned out to be dependent on visceral adiposity [199,200]. Another supporting observation study conducted by Li et al. revealed a positive association between VAT and MFI together with both endoscopic scores and disease recurrence [201]. Notably, the VAT area above the median value was described as a predictor of clinical CD recurrence after surgery, too. How about dosing oral drugs in the course of IBD considering the molecular patterns connected with nutritional status, adipose tissue, and changes in body mass of the patients? Azathioprine, a crucial agent in chronic immunosuppressive therapy, is dosed according to the weight, and its serum concentration is not routinely measured in the situation of achieved response to the therapy. However, according to past studies, clinical remission achieved with azathioprine correlates with the concentration of its derivative in the blood—6-thioguanine nucleotide (6-TGN) [202]. Nevertheless, further looking for a potential dependency between the content of subcutaneous or visceral fat and therapeutic levels of 6-TGN did not bring any significant data. Thus, treatment with thiopurines in IBD was shown to be unrelated to the distribution of fat [203].

## 16. Biologics, Obesity, and IBD—What Do We Know?

On the contrary, the issue of biological treatment might constitute here a more challenging task. The data concerning the impact of obesity on the response to biologics in the course of IBD are quite scanty. Some of the biologics are dosed in a weight-adjusted manner. Higher body weight can be associated with an increased volume of distribution of certain monoclonal antibodies, e.g., infliximab administration. Regarding the anti-TNF-α agents, it is worth mentioning that their administration in the course of autoimmune diseases (other than IBD) to patients with a higher BMI produced a worse clinical response. This pattern was confirmed in surveys conducted in the course of IBD as well. Nonetheless, investigations conducted on VDZ have not presented any advantage of weight-adjusted dosing over fixed doses of the drug. The study performed on IBD patients treated with VDZ to find out if there is any significant role of obesity in the context of treatment revealed that being overweight did not stand for a factor connected with the need for VDZ dose escalation or achievement of steroids-free remission. Nonetheless, the normalization of CRP in the examined group occurred more often in non-obese patients, even though the results concerning endoscopic remission observed during the treatment were not associated with the presence of obesity [204,205]. A study performed on 160 patients with UC treated with biologics demonstrated that an increase in the BMI by each 1 kg/m^2^ was related to an increased risk of treatment failure and surgery/hospitalization of up to 4% and 8%, respectively [206]. Nevertheless, an analysis performed on a population of 1206 subjects with IBD (CD and UC) did not show any differences according to the character of response to treatment in obese and nonobese patients treated with infliximab [207]. One should remember that the BMI does not serve as a tool for estimating the volume of subcutaneous and visceral adipose tissue, so its application does not provide an accurate assessment of the direct relationship between adipose tissue and clinical presentation of IBD. Thus, it is worth mentioning investigations that concerned the evaluation of the dependency between fat distribution and the results of biological treatment. Shen et al. evaluated CD patients during an infliximab induction therapy, noting a significant association between visceral (but not subcutaneous) fat adiposity assessed with the CT-derived MFI and the degree of mucosal healing [208,209]. Taking into consideration former surveys, it becomes quite clear that the overall response to biologics in obese IBD patients is worse. Furthermore, obesity is directly linked to a more severe course of the disease, a greater tendency to relapse, and a generally poorer quality of everyday life. The aspect of visceral adipose tissue can not be neglected either. It is known as the source of proinflammatory molecules which can directly interact with biological therapies. Independently of the manner of drug exposure, elevated BMI turned out to be associated with treatment failure and an increased probability of hospitalization together with surgical treatment. Simultaneously, obese patients were shown to require a more frequent biologic dose escalation compared to normal-weight patients.

## 17. Body Mass and Its Clinical Value in the Management of IBD Patients

Interestingly, there is a lack of trustworthy data in clinical trials on possible attempts to normalize the weight of obese IBD patients in order to look for potential improvements in their outcomes. It might be relatively emerging information as so far the clinical character of IBD measured with BMI has been observed to be quite significantly related to the context of outcome or disease severity. The BMI > 30 kg/m^2^ has been shown to be connected with a greater number of perianal complications and hospitalizations in the course of IBD. Another observation linked obesity in CD patients with relatively earlier surgeries and their worse outcomes, even in the pediatric IBD population. According to the manifestations of the disease, extraintestinal symptoms of IBD have even been linked to higher levels of the BMI. However, one should be careful with the final evaluation of this issue because other investigations have not revealed a certain association between the BMI and the course of IBD. However, a potential linkage between obesity, quality of life, and healthcare expenses related to obese IBD patients can not be omitted all in all. Additionally, it was pointed out several times that a linear relationship between the BMI and the total body fat is definitely unsatisfactory, indicating that the focus on general body fat distribution could be more relevant compared to overall obesity. To confirm this suggestion, it is worth emphasizing that visceral adiposity in CD patients is a more trustworthy marker of obesity connected with the risk of IBD-followed complications than the BMI. VAT, MFI, or VAT/subcutaneous adipose tissue (SAT) ratio obtained via computed tomography (CT) or dual-energy X-ray absorptiometry (DXA) were described as valuable tools in the assessment of postoperative recurrence or progression in the course of CD [163,210].

## 18. IBD and Anti-Inflammatory Diets—Are There any Reliable Changes in Front of Us?

Nowadays, more and more theories focus on the idea of an anti-inflammatory diet that could be implemented in various pathologies directly linked to the inflammation in their background. The concept of a dietary inflammation index (DII) was invented to make it possible to verify the direct impact of the pattern of a whole diet on the content of inflammation particles circulating in the bloodstream. Due to the DII, there are diets with pro- and anti-inflammatory properties. According to this division, the Nordic, Mediterranean, washoku, and Jiangnan diets have been shown to exert anti-inflammatory action. On the contrary, the so-called Western diet, rich in, e.g., refined grains and sugars, salt, processed meat, and high-fat products, promotes inflammation, resulting finally in the extra buildup of visceral adipose tissue with the induction of the release of proinflammatory adipokines. The Western diet is also a known source of the advanced glycation end products (AGEs) involved in the development of age-associated diseases [211,212]. Going further, obesity influences the microbiome, reversing the proportion between Bacteroidetes and Firmicutes, leading finally to the increase in Firmicutes up to 50%. In such circumstances, the microbiome is able to generate energy from food to a greater extent compared to lean subjects. A common phenomenon, due to dysbiosis, observed simultaneously in obesity and IBD concerns a modified ratio between obligate and facultative anaerobes, favoring their second type. Elevated concentrations of both *E. coli* and *R. gnavus* observed in clinical studies confirm this point of view [213,214,215]. Moving back to the general idea of malnutrition among IBD patients, it is prevalent in a significant number of them, ranging from 20% to 85%, according to multiple observations, reaching the highest values in hospitalized CD patients. Simultaneously, 15–40% of IBD adults suffer from obesity; the remaining 20–40% are overweight, with similar proportions between CD and UC. One should not forget about the possibility of accompanying sarcopenia. Thus, sarcopenic obesity might become a problem to face, with its negative influence on the general outcome of treatment and the quality of patients’ life [216]. The aforementioned abnormalities highlight the potential impact of the clinical dependences between the microbiome, nutritional status, and coexisting inflammation in the natural history of IBD. They interact directly with adipose tissue and might interfere with certain adipokines. All in all, there is still a great demand for novel nutritional trials among IBD patients to settle the role of obesity in the spectrum of pharmacological treatment and elucidate in detail potential benefits to the management due to the introduction of changes in everyday diet. Notably, some data concern the possible role of a low fermentable oligosaccharides, disaccharides, monosaccharides, and polyols (FODMAP) diet in the IBD setting [217]. Its introduction for six weeks was connected with lowering the concentration of fecal calprotectin and improved quality of life in examined patients (probably due to the attenuation of functional symptoms, e.g., bloating and flatulence). In a parallel study, fructans were shown to exacerbate the abovementioned complaints. Another study performed among IBD patients on a low FODMAP diet revealed a decrease in the variety of fecal microbiota capable of modulating the immune response, with no relevant impact on the parameters of inflammation [218,219,220]. Recent reports have indicated that a low FODMAP diet is capable of improving functional gastrointestinal symptoms in IBD subjects with no significant improvement in stool consistency and mucosal healing. Thus, it might improve the quality of life of patients, but its role in fighting mucosal inflammation is questionable [221,222].

## 19. Novel Potential Options of Treatment in IBD Patients from the Perspective of Adipose Tissue

Up to now, we do not have any commonly available therapeutic strategies targeted specifically at visceral or mesenteric fat. Considering the potential molecular pathways to be blocked, from the pharmacological point of view, PPAR-γ should be explored as a potential molecule to be captured. It is involved in adipocyte hyperplasia within MAT and presents increased activation in CD patients compared to the controls [223,224,225]. Upregulation of PPAR-γ has quite a complex background, being related to obesity, high dietary intake of fatty acids, or followed by the stimulation of TLR4 due to bacterial metabolites [226]. The abovementioned phenomena may result in the formation of creeping fat. Acting against PPAR-γ could be perceived as a rational possibility for the treatment. The issue is not so obvious, unfortunately, as PPAR-γ signaling is involved in the epithelial expression of beta-defensin *DEFB1*, too. This molecule protects against mucosal adherence of certain microorganisms; its expression has been noted to be decreased in colonic CD. On the contrary, PPAR-γ signaling in UC patients seems to be altered and is associated negatively with the endoscopic manifestation of the disease [227]. Experimental colitis treated with PPAR-γ agonists improved the pattern of tissue histology [228]. In patients with UC, the administration of rosiglitazone (a PPAR-γ agonist) improved the outcome [229]. Nevertheless, safety aspects connected with the intake of this drug may provoke some doubts [230]. Mesalazine, commonly known and used in the first line of therapy in UC, can also behave as a ligand for PPAR-γ. Moreover, there are ongoing surveys concerning the invention of new forms of 5-aminosalicylic acid (5-ASA) with even greater affinity for PPAR-γ [231]. GED-0507-34 Levo is a potential brand-new PPAR-γ modulator exhibiting relatively good effects in the improvement of colitis and intestinal fibrosis found in recent studies [232].

## 20. Conclusions

The molecular background of the involvement of obesity, adipose tissue with its reactive products, and underlying inflammation in the course of IBD belong to the areas of current global interest in gastroenterology. Everything is due to the complexity of this phenomenon and the potential effect of visceral and mesenteric fat exerted on the clinics in the natural history of IBD (e.g., treatment, prediction of remission, and overall management). We are probably about to uncover a missing element in the treatment of IBD and make it more individualized. Nevertheless, there is a lot of future surveys to be conducted, translating molecular findings into everyday clinical practice in this context. They should especially concern the potential role of adipokines as a target for future treatment. Metabolic disorders should also be examined together with coexisting IBD because of the common phenomena in their pathogenesis. Finally, obesity (even though its prevalence in IBD subjects seems to be similar to healthy population) must be treated as an additional disorder of IBD patients that might have a direct impact on the treatment response. Thus, the take-home message is best stated as follows: because of its complex role in the course of IBD, adipose tissue constitutes an independent inflammatory organ; therefore, its detailed exploration may revolutionize the way of therapeutic thinking and management of IBD patients in the near future.

## Figures and Tables

**Figure 1 nutrients-14-03983-f001:**
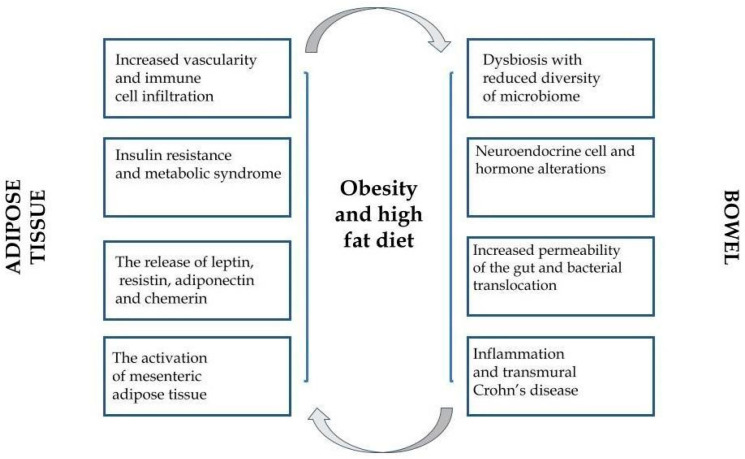
The interactions between obesity, adipose tissue, and inflammatory bowel disease.

## Data Availability

All the data are included in the manuscript.

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
