# Peer review of "Impact of Obesity on the Course of Management of Inflammatory Bowel Disease—A Review"

_nutrients, 2022, doi:10.3390/nu14193983_

Round 1
Reviewer 1 Report
Molecular background of the involvement of obesity, adipose tissue with its reactive products and inflammation in the course of IBD is currently one of the most frequently studied topics in the etiopathogenesis, course and treatment of IBD. Recently, several studies have been published indicating a direct impact of body fat on the effects of pharmacological treatment in IBD.
The authors of the submitted manuscript analyzed publications regarding IBD in the context of obesity and associated disorders, focusing on adipokines, creeping fat and possible relationships between these disorders and the treatment of IBD patients. This subject is a novel research direction in IBD pathogenesis and treatment.
The authors of the manuscript “IMPACT OF OBESITY ON THE COURSE OF MANAGEMENT OF INFLAMMATORY BOWEL DISEASE”, present a very thoughtful analysis of the latest data, published mainly in the period 2019-2022.
I have no comments or objections to the reviewed manuscript.
Reviewer 2 Report
Overall, this is an interesting review of the literature exploring the association between obesity and its comorbidities with IBD. The review provided a general overview on several different topics (mostly various adipokines), but the relevance to the topic of the review (obesity & IBD) was not clear. There should be more effort to maintain the focus of the review on the topic at hand. The placement of the in-text citations was not thorough and left the reader wondering which reference was being referred to in each sentence. Suggestions for clarifying the references can be found below, in addition to other recommendations for improving this review.
The paper could benefit from proofreading for English grammar and syntax. While overall, the paper is well-written, there is phrasing throughout that not consistent with typical English scientific phraseology.
Line 33-34: How does the prevalence of obesity among adults with UC or CD differ from the general population? Meaning, are people with UC or CD more likely to have obesity than someone without these conditions? The prevalence of obesity of 15% to 40% is similar to the prevalence of obesity in the general adult population.
· Line 36-37: please provide a reference (or references) for this statement: “Obesity can be also linked to a higher risk of the development of CD; however this association seems to be unrelated to UC.”
· Line 38-40: Please clarify if maternal obesity is linked to increased risk of CD in the mother or the offspring.
· Line 62-64: please cite the cohort study referred to in this sentence.
· Line 64-67: please cite the European Prospective Investigation into Cancer and Nutrition that is mentioned in this sentence.
· Line 79-81: Please cite the multiple reports you refer to in this sentence: “Multiple reports indicate a higher prevalence of the coexistence…”.
· Line 85-86: Please provide a reference in support of this statement “…sharing common genetic variants was raised as a link between UC and diabetes.”
· Line 86-88: please provide a reference in support of this statement “Moreover, the observation of comorbidity of IBD and type 2 diabetes in former surveys highlighted intestinal dysbiosis and altered epithelial barrier as proposed causative factors.”
· Line 120-121: Please provide references in support of this statement “Investigations conducted on colonic tissue in mice and humans diagnosed with colitis/IBD revealed an increased concentration of apelin.”
· Line 122-123: Please provide a reference in support of this statement “Of note, adding the synthetic apelin to the cell culture turned out to stimulate the proliferation of epithelial cells.”
· Line 140-141: “testosterone” and “hormones” are misspelled. Also, please provide a reference in support of this statement about testosterone and ovarian hormones regulating leptin production and that leptin concentrations differ between sexes.
· Line 148-149: The authors state that low leptin concentrations are commonly observed in obesity. This is not the case. Serum leptin concentrations circulate in proportion to body mass. There a few, extremely rare cases of congenital leptin deficiency, which leads to obesity. However, the majority of people with obesity have high leptin concentrations.
· Line 170-172. Please provide a reference in support of this statement “Notably, Treg lymphocytes constitute an essential source of leptin, simultaneously secreting and presenting receptors for this adipokine.”
· Line 179-180: Please cite the De Rosa paper you reference in this sentence.
· At times the review seems to stray away from its primary focus. For example, section #5 goes into great detail on leptin concentrations in different sexes, leptin’s physiologic roles in energy balance, exogenous leptin treatment, etc. This seems like a great deal of extraneous information unrelated to the obesity-IBD link. It is recommended that this section be streamlined to maintain focus on the topic to be reviewed (or at least link what is covered in the leptin section to obesity & IBD). The most important part of section 5, as it relates to the topic of the review, are the last few sentences from line 200-206 that directly relate increased concentrations of leptin (a hormone elevated in obesity) in UC and CD patients. These sentences also link leptin to inflammatory factors and mesenteric adipose tissue of CD and UC patients. These last few sentences of section five should be expanded upon and should make up the bulk of that section. There should be less time spent on general functions of leptin, and more time spent on leptin and inflammation, the intestine, and IBD.
· The same comment as above applies to section #6 about adiponectin. The readers do not need such a long overview of general functions of adiponectin (circadian rhythm, sex differences, PEPCK, glycolysis, PPARs in the liver, etc. How does this relate to obesity & IBD, which is the focus of the review?). The reader will be interested in how adiponectin may mediate the link between obesity and IBD. The authors should maintain focus on this topic in this section.
· Lines 257-260. This sentence is not clear. This sentence needs to be reworded in order to understand its meaning “Nishihara et al. followed a similar manner of the investigation and finally, due to the achieved results, complicated a previous point of view; namely adding the adiponectin constituted an anti-inflammatory factor; perhaps because of the direct stimulation of colonic epithelial cells.” The sentence immediately following this sentence is also unclear. I don’t understand what is meant by either of these sentences.
· Line 280-281. Please cite the 2001 paper referred to in this sentence “2001 appears to be the first time when resistin was described as a peptide released by adipocytes, responsible for the development of insulin resistance.”
· The relevance of Section #7 on resistin to the topic of the review (obesity & IBD) needs to be made clearer. This section is basically an overview of resistin’s role in inflammation, atherosclerosis, bone turnover, liver failure, kidney function, etc. The section should retain focus on the role of resistin in obesity and how this relates to IBD. There is only one sentence about this (the last sentence of the paragraph). That sentence should be expanded upon and should be the focus of that section.
· Line 325: “chemerin” is misspelled.
· Section #10 should be broken up into multiple paragraphs.
· Lines 383-385. Please provide a reference for this statement “Interestingly, some data indicate that MAT can exert a protective action due to the improvement of local host defense by stimulating local inflammation, which can even reduce the risk of perforation.”.
· Lines 385-388. Please provide a reference for this statement “According to this point of view, the expression of leptin (enhancing the expression of toll-like receptors (TLRs) in pre-adipocytes and their mature forms), has been shown to be upregulated locally, but not systemically in the course of IBD”.
· Line 449: No need to abbreviate “creeping fat” since it was referred to ~10 times before this abbreviation, and the abbreviation is not used elsewhere in the text.
· Section #12 should be broken up into multiple paragraphs.
· Line 580 “Mediterranean” is misspelled.
· Line 586-600: There is a great deal of information that appears to unrelated to the section topic, which is anti-inflammatory diets for IBD. These lines discuss microbiota, malnutrition, sarcopenia, etc. Unless the relevance of this information to anti-inflammatory diets for IBD is made clear, it should be removed.
Section 13: There is a great deal of information on the FODMAP diet for IBD, and this was very superficially covered. Unless a thorough and accurate review of the literature on anti-inflammatory diets for IBD is done, the authors may want to consider removing section 13.
· The Conclusions section is unclear. The wording of the sentences is not clear, and I am not left with a take home message or “conclusion” of the literature review I just read. In the Conclusions section the authors basically state that the role of obesity and adipose tissue in IBD is of interest and more studies need to be done. That is not much of a conclusion. Can the authors clarify in a few sentences what the take home message of the review is? Can the authors specific clearly what areas need to be studied?
· A figure summarizing the proposed link between obesity, the adipokines discussed, and IBD would strengthen the message of the review.
· Line 369 starting with “Bowel inflammation – Line 382 are taken from section 3.2 of PMID 33921758. This paragraph is taken nearly verbatim from the source with minor changes (inserting synonyms).
Round 2
Reviewer 2 Report
Line 36-37: please provide a reference (or references) for this statement: “Obesity can be also linked to a higher risk of the development of CD; however this association seems to be unrelated to UC. References were provided in the response to reviewer. These references need to go into the manuscript.
Line 38-40: Please clarify if maternal obesity is linked to increased risk of CD in the mother or the offspring. This was not clarified.
Moreover, according to the Danish National Birth Cohort (counting more than 75 thousands women), obesity before getting pregnant was connected with 1,9-fold increased risk of CD, (but not UC) in the offspring?
Line 86-88: please provide a reference in support of this statement “Moreover, the observation of comorbidity of IBD and type 2 diabetes in former surveys highlighted intestinal dysbiosis and altered epithelial barrier as proposed causative factors.” A reference was provided in the response to reviewer. This needs to go into the manuscript.
I asked for a citation of the De Rosa paper that was mentioned in the manuscript. In the manuscript it is stated "De Rosa with co-workers...". A citation has since been added, but this citation has no author with the name "De Rosa". Please clarify.
Author Response
"Please see the attachment."
